# SIMPLIFYING REFERRED VISUAL SEARCH WITH CONDITIONAL CONTRASTIVE LEARNING

## ABSTRACT

This paper introduces a new challenge for image similarity search in the context of fashion, addressing the inherent ambiguity in this domain stemming from complex images. We present Referred Visual Search (RVS), a task allowing users to define more precisely the desired similarity, following recent interest in the industry. We release a new large public dataset, LAION-RVS-Fashion, consisting of 272k fashion products with 842k images extracted from LAION, designed explicitly for this task. However, unlike traditional visual search methods in the industry, we demonstrate that superior performance can be achieved by bypassing explicit object detection and adopting weakly-supervised conditional contrastive learning on image tuples. Our method is lightweight and demonstrates robustness, reaching Recall at one superior to strong detection-based baselines against 2M distractors. Code, data, and models will be released.

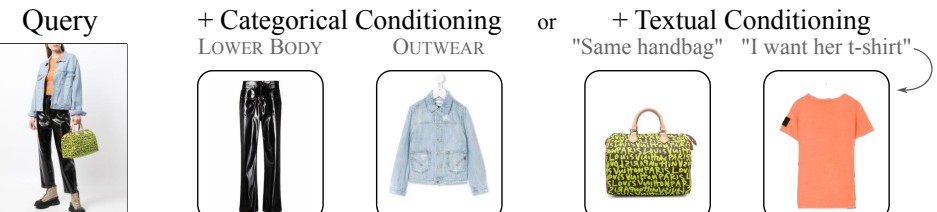

Figure 1: Overview of the Referred Visual Search task. Given a query image and conditioning information, the goal is to retrieve a target instance from among a large number of distractors.

## 1 INTRODUCTION

Image embeddings generated by deep neural networks play a crucial role in a wide range of computer vision tasks. Image retrieval, has gained substantial prominence, leading to the development of dedicated vector database systems (Johnson et al., 2019). These systems facilitate efficient retrieval by comparing embedding values and identifying the most similar images within the database.

Image similarity search in the context of fashion presents a unique challenge due to the inherently ill-founded nature of the problem. The primary issue arises from the fact that two images can be considered similar in various ways, leading to ambiguity in defining a single similarity metric. For instance, two images of clothing items may be deemed similar based on their color, pattern, style, or even the model pictured. This multifaceted nature of similarity in fashion images complicates the task of developing a universally applicable similarity search algorithm, as it must account for the various ways in which images can be related.

An intuitive approach is to request users furnish supplementary information delineating their interests, such as providing an image of an individual and denoting interest in the hat (see Fig. 1). Numerous industry leaders including Google, Amazon, and Pinterest have adopted this tactic, however academic discourse on potential alternative methodologies for this task remains scarce. For convenience, we propose terming this task Referred Visual Search (RVS), as it is likely to garner attention from the computer vision community due to the utility for product search in extensive catalogs.

In practice, object selection in complex scenes is classically tackled using object detection and crops (Jing et al., 2015; Hu et al., 2018; Ge et al., 2019; Shiau et al., 2020). Some recent approaches use categorical attributes (Dong et al., 2021) or text instead (Das et al., 2022), and automatically crop the image based on learned attention to input attributes. Indeed it is also possible to ask the user to perform the crop himself, yet in all the situations the performance of the retrieval will be sensitive to this extraction step making it costly to build a generic retrieval tool. Recently, Jiao et al. (2023) went a step further, incorporating prior knowledge about the taxonomy of fashion attributes and classes without using crops. They use a multi-granularity loss and two sub-networks to learn attribute and class-specific representations, resulting in improved robustness for fashion retrieval, yet without providing any code. In this work, we seek to advance in this direction and totally eliminate the need for explicit detection or segmentation while still producing similarities in the embedding space specific to the conditioning. Indeed, all localization-dependent approaches hinge on multi-stage processing heuristics particular to the dataset, whereas an end-to-end approach has the potential to be more generalizable and robust. To our knowledge, no public dataset is available for this task. Therefore, to demonstrate our approach's soundness, we extracted a subset of LAION 5B focused on pairs of images sharing a labeled similarity in the domain of Fashion.

This paper presents two contributions to the emerging field of Referred Visual Search, aiming at defining image similarity based on conditioning information.

- ✓ The introduction of a new dataset, referred to as LAION-RVS-Fashion, which is derived from the LAION-5B dataset and comprises 272k fashion products with nearly 842k images. This dataset features a test set with an addition of more than 2M distractors, enabling the evaluation of method robustness in relation to gallery size. The dataset's pairs and additional metadata are designed to necessitate the extraction of particular features from complex images.
- ✓ An innovative method for learning to extract referred embeddings using weakly-supervised training. Our approach demonstrates superior accuracy against a strong detection-based baseline and existing published work. Furthermore, our method exhibits robustness against a large number of distractors, maintaining high R@1 even when increasing the number of distractors to 2M.

## 2 RELATED WORK

**Multi-modal Models.** Deep learning has made significant progress in both vision and language domains, leading to the emergence of new vision-language methods. This new field notably developed Vision-Language Pre-Training (Du et al., 2022b), leveraging many pretext tasks to create models that can be finetuned for downstream multi-modal applications. Specific models have been trained on fashion datasets to extract more relevant features (Zhuge et al., 2021; Mirchandani et al., 2022; Goenka et al., 2022; Ji et al., 2023), and applied to multi-modal product retrieval (Zhan et al., 2021; Yu et al., 2022; Zheng et al., 2023). In our work, we use CLIP (Radford et al., 2021) as a general feature extractor for our visual encoder.

Vision-Language processing also brought new challenges, in particular Referring Expression Comprehension and Segmentation where a sentence designates an object in a scene, that the network has to localize. For the comprehension task (similar to open-vocabulary object detection) the goal is to output a bounding box (Luo et al., 2020; Zeng et al., 2022a;b; Liu et al., 2023). The segmentation task aims at producing an instance mask for images (Zhang et al., 2017; Luo et al., 2020; Huang et al., 2020; Ding et al., 2021; Kirillov et al., 2023) and recently videos (Wu et al., 2022; Botach et al., 2022). In this paper, we propose a referring expression task, where the goal is to embed the designated object of interest into a representation that can be used for retrieval. We explore the use of Grounding DINO (Liu et al., 2023) and Segment Anything (Kirillov et al., 2023) to create a strong baseline on our task.

**Instance Retrieval.** In the last decade, content-based image retrieval has changed because of the arrival of deep learning, which replaced many handcrafted heuristics (keypoint extraction, descriptors, geometric matching, re-ranking...) (Dubey, 2022). In the industry this technology has been of interest to retail companies and search engines to develop visual search solutions, with new challenges stemming from the large scale of such databases. Initially using generic pretrained backbones to extract embeddings with minimal retraining (Yang et al., 2017), methods have evolved toward domain-specific embeddings supervised by semantic labels, and then multi-task domain-specific embeddings,

leveraging additional product informations (Zhai et al., 2019; Bell et al., 2020; Tran et al., 2019). The latest developments in the field incorporate multi-modal features for text-image matching (Zhan et al., 2021; Yu et al., 2022; Zheng et al., 2023), with specific vision-language pretext tasks.

However, these methods often consider that the query image is unambiguous, and often rely on a region proposal system to crop the initial image (Jing et al., 2015; Zhang et al., 2018; Hu et al., 2018; Shiau et al., 2020; Bell et al., 2020; Du et al., 2022a). In our work, we bypass this step and propose an end-to-end framework, leveraging the Transformer architecture to implicitly perform this detection step conditionally to the referring information.

**Conditional Embeddings.** Conditional similarity search has been attempted in two ways: by modifying the retrieval process or the embedding process. On one hand, for the retrieval process, Hamilton et al. (2021) propose to use a dynamically pruned random projection tree. On the other hand, for the embeddings, previous work in conditional visual similarity learning has been oriented toward attribute-specific retrieval, considering that different similarity spaces should be defined depending on chosen discriminative attributes (Veit et al., 2017; Mu & Guttag, 2022). These approaches use either a mask applied on the features (Veit et al., 2017), or different projection heads (Mu & Guttag, 2022), and require extensive data labeling.

In Fashion, ASEN (Ma et al., 2020) uses spatial and channel attention to an attribute embedding to extract specific features in a global branch. Dong et al. (2021) and Das et al. (2022) build upon this model and add a local branch working on an attention-based crop. Recently, Jiao et al. (2023) incorporated prior knowledge about fashion taxonomy in this process to create class-conditional embeddings based on known fine-grained attributes, using multiple attribute-conditional attention modules. In a different domain, Asai et al. (2022) tackle a conditional document retrieval task, where the user intent is made explicit by concatenating instructions to the query documents. In our work, we use Vision Transformers (Dosovitskiy et al., 2021) to implicitly pool features depending on the conditioning information, without relying on explicit ROI cropping or labeled fine-grained attributes.

**Composed Image Retrieval.** Composed Image Retrieval (CIR) (Vo et al., 2019) is another retrieval task where the embedding of an image must be modified following a given instruction. Recent methods use a composer network after embedding the image and the modifying text (Lee et al., 2021; Chen et al., 2022; Baldrati et al., 2023). While CIR shares similarities with RVS in terms of inputs and outputs, it differs conceptually. Our task focuses on retrieving items based on depicted attributes and specifying a similarity computation method, rather than modifying the image.

In Fashion, CIR has been extended to dialog-based interactive retrieval, where an image query is iteratively refined following user instructions (Guo et al., 2018; Wu et al., 2019; Yuan & Lam, 2021; Han et al., 2022).

**Retrieval Datasets.** Standard datasets in metric learning literature consider that the images are object-centric, and focus on single salient objects (Wah et al., 2011; Krause et al., 2013; Song et al., 2016). In the fashion domain there exist multiple datasets dedicated to product retrieval, with paired images depicting the same product and additional labeled attributes. A recurrent focus of such datasets is cross-domain retrieval, where the goal is to retrieve images of a given product taken in different situations, for exemple consumer-to-shop (Liu et al., 2012; Wang et al., 2016; Liu et al., 2016; Ge et al., 2019), or studio-to-shop (Liu et al., 2016; Lasserre et al., 2018). The domain gap is in itself a challenge, with issues stemming from irregular lighting, occlusions, viewpoints, or distracting backgrounds. However, the query domain (consumer images for exemple) often contains scenes with multiple objects, making queries ambiguous. This issue has been circumvented with the use of object detectors and landmarks detectors (Kiapour et al., 2015; Huang et al., 2015; Liu et al., 2016; Ge et al., 2019). (Kiapour et al., 2015; Liu et al., 2016; Wang et al., 2016) are not accessible anymore.

With more than 272k distinct training product identities captured in multi-instance scenes, our new dataset proposes an exact matching task similar to the private Zalando dataset (Lasserre et al., 2018), while being larger than existing fashion retrieval datasets and publicly available. We also create an opportunity for new multi-modal approaches, with captions referring to the product of interest in each complex image, and for robustness to gallery size with 2M added distractors at test time.

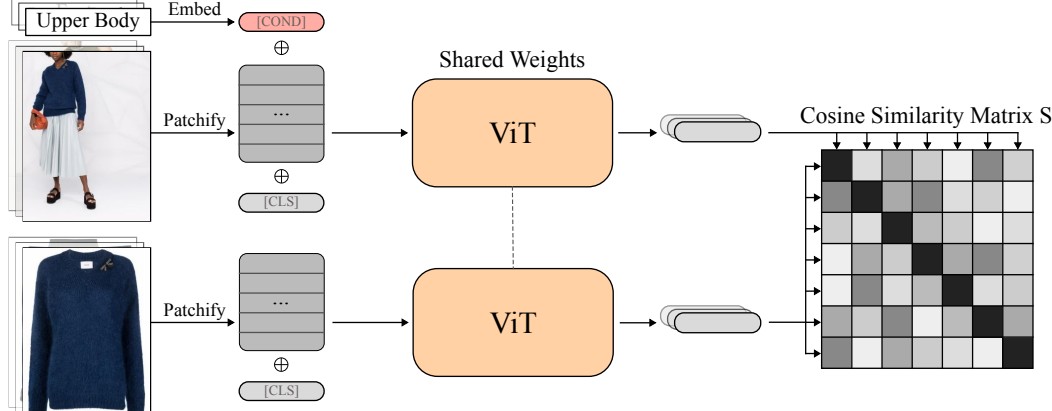

Figure 2: Overview of our method on LRVS-F. For each element in a batch, we embed the scene conditionally and the isolated item unconditionally. We optimize a Normalized Temperature-scaled Cross Entropy loss over the cosine similarity matrix. ⊕ denotes concatenation to the patch sequence.

## 3   CONDITIONAL EMBEDDING

**Task Formulation:**   Let $x_q$ be a query image, and $c_q$ associated referring information. Similarly, let $x_t$ be a target image, described by the latent information $c_t$. Both $c_q$ and $c_t$ can be thought of as categories or textual referring information. The probability of $x_t$ to be relevant for the query $x_q$ is given by the conditional probability $P(x_t, c_t | x_q, c_q)$. When working with categories, a filtering strategy consists in assuming independence between the images and their category,

$$P(x_t, c_t | x_q, c_q) = P(x_t | x_q) P(c_t | c_q) , \qquad (1)$$

and further assuming that categories are uncorrelated (*i.e.*, $P(c_t | c_q) = \delta_{c_q = c_t}$ with $\delta$ the Dirac distribution). In this work, we remove those assumptions and instead assume that $P(x_t, c_t | x_q, c_q)$ can be directly inferred by a deep neural network model. More specifically, we propose to learn a flexible embedding function $\phi$ such that

$$\langle \phi(x_q, c_q), \phi(x_t, c_t) \rangle \propto P(x_t, c_t | x_q, c_q) . \qquad (2)$$

Our approach offers a significant advantage by allowing the flexibility to change the conditioning information ($c_q$) at query time, enabling a focus on different aspects of the image.

**Method:**   We implement $\phi$ by modifying the Vision Transformer (ViT) architecture (Dosovitskiy et al., 2021). The conditioning is an additional input token with an associated learnable positional encoding, concatenated to the sequence of image patches. The content of this token can either be learned directly (*e.g.* for discrete categorical conditioning), or be generated by another network (*e.g.* for textual conditioning). We experimented with concatenating at different layers in the transformer, and found that concatenating before the first layer is the most sensible choice (see Appendix B.3). At the end of the network, we linearly project the [CLS] token to map the features to a metric space.

We compute the similarity between two embeddings $z_i = \phi(x_i, c_i), z_j = \phi(x_j, c_j) \in \mathbb{R}^d$ with the cosine similarity $s(z_i, z_j) = z_i^\top z_j / (\|z_i\| \|z_j\|)$. In practice we normalize the embeddings to the hypersphere at the end of the network, and use simple inner products during training and retrieval.

We train the network with Normalized Temperature-scaled Cross Entropy Loss (NT-Xent) (Sohn, 2016; Chen et al., 2020), using the same variation as CLIP (Radford et al., 2021), which is detailed in the next paragraph. However, we hypothesize that even though our method relies on a contrastive loss, it does not explicitly require a specific formulation of it. We choose the NT-Xent loss because of its popularity and scalability.

During training, given a batch of $N$ pairs of images and conditioning $((x_i^A, c_i^A); (x_i^B, c_i^B))_{i=1..N}$, we compute their conditional embeddings $(z_i^A, z_i^B)_{i=1..N}$, and a similarity matrix $S$ where $S_{ij} =$

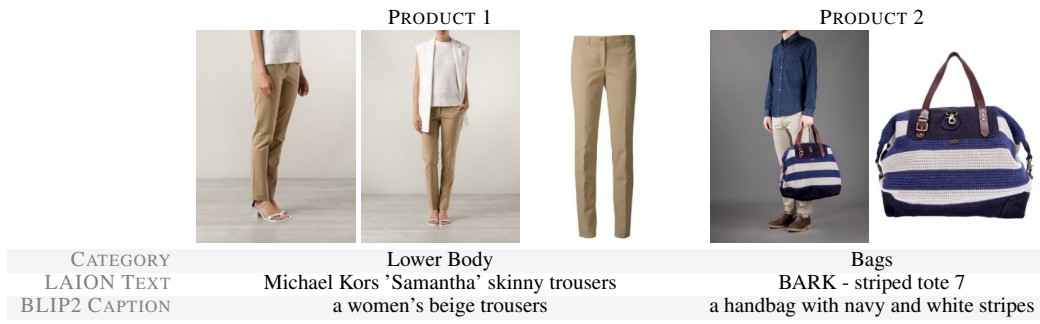

| | PRODUCT 1 | PRODUCT 2 |
|---|---|---|
| CATEGORY | Lower Body | Bags |
| LAION TEXT | Michael Kors 'Samantha' skinny trousers | BARK - striped tote 7 |
| BLIP2 CAPTION | a women's beige trousers | a handbag with navy and white stripes |

Figure 3: Samples from LRVS-F. Each product is represented on at least a simple and a complex image, and is associated with a category. The simple images are also described by captions from LAION and BLIP2. Please refer to Appendix A.1 for more samples.

$s(z_i^A, z_j^B)$. We then optimize the cosine similarity of the correct pair with a cross-entropy loss, effectively considering the $N-1$ other products in the batch as negatives:

$$l(S) = -\frac{1}{N} \sum_{i=1}^{N} \log \frac{\exp(S_{ii}\tau)}{\sum_{j=1}^{N} \exp(S_{ij}\tau)} , \qquad (3)$$

with $\tau$ a learned temperature parameter, and the final loss is $\mathcal{L} = l(S)/2 + l(S^\top)/2$. Please refer to Fig. 2 for an overview of the method.

At test time, we use FAISS (Johnson et al., 2019) to create a unique index for the entire gallery and perform fast similarity search on GPUs.

## 4 DATASET

Metric learning methods work by extracting features that pull together images labeled as similar (Dubey, 2022). In our case, we wanted to create a dataset where this embedding has to focus on a specific object in a scene to succeed. We found such images in fashion, thanks to a standard practice in this field consisting in taking pictures of the products alone on neutral backgrounds, and worn by models in scenes involving other clothing items (see Fig. 3).

We created LAION-RVS-Fashion (abbreviated LRVS-F) from LAION-5B by collecting images of products isolated and in context, which we respectively call *simple* and *complex*. We grouped them using extracted product identifiers. We also gathered and created a set of metadata to be used as referring information, namely LAION captions, generated captions, and generated item categories. Please refer to Appendix A.2, A.3 and C for metadata details and a datasheet (Gebru et al., 2021).

In total, we extracted 272,451 products for training, represented in 841,718 images. This represents 581,526 potential simple/complex positive pairs. We additionally extracted 400 products (800 images) to create a validation set, and 2,000 products (4,000 images) for a test set. We added 99,541 simple images in the validation gallery as distractors, and 2,000,014 in the test gallery. Details about the benchmark will be given in section 4.2.

### 4.1 CONSTRUCTION

**Image Collection.** The URLs in LRVS-F are a subset of LAION-5B, curated from content delivery networks of fashion brands and retailers. By analyzing the URL structures we identified product identifiers, which we extracted with regular expressions to recreate groups of images depicting the same product. URLs without distinct identifiers or group membership were retained as distractors.

**Annotations.** We generated synthetic labels for the image complexity, the category of the product, and added new captions to replace the noisy LAION alt-texts. For the complexity labels, we employed active learning to incrementally train a classifier to discern between isolated objects on neutral backdrops and photoshoot scenes. The product categories were formed by aggregating various

fine-grained apparel items into 10 coarse groupings. This categorization followed the same active learning protocol. Furthermore, the original LAION captions exhibited excessive noise, including partial translations or raw product identifiers. Therefore, we utilized BLIP-2 (Li et al., 2023) to generate new, more descriptive captions.

We randomly sampled images and manually verified the quality of the labels. For the complexity labels, we measured an empirical error rate of $1/1000$ on the training set and $3/1000$ for the distractors. For the product categories, we measured a global empirical error rate of $1\%$, with confusions mostly arising from semantically similar categories and images where object scale was ambiguous in isolated settings (e.g. long shirt vs. short dress, wristband vs. hairband). The BLIP2 captions we provided exhibit good quality, increasing the mean CLIP similarity with the image by $+7.4\%$. However, as synthetic captions, they are not perfect and may contain occasional hallucinations.

**Dataset Cleaning.** In order to mitigate false negatives in our results, we utilized Locally Sensitive Hashing and OpenCLIP ViT-B/16 embeddings to eliminate duplicates. Specifically, we removed duplicates between the test targets and test distractors, as well as between the validation targets and validation distractors. Throughout our experiments, we did not observe any false negatives in the results. However, there remains a small quantity of near-duplicates among the distractor images.

## 4.2 BENCHMARK

We define a benchmark on LRVS-F to evaluate different methods on a held-out test set with a large number of distractors. The test set contains 2,000 unseen products, and up to 2M distractors. Each product in the set is represented by a pair of images - a simple one and a complex one. The objective of the retrieval task is to retrieve the simple image of each product from among a vast number of distractors and other simple test images, given the complex image and conditioning information.

For this dataset, we propose to frame the benchmark as an asymmetric task : the representation of simple images (the gallery) should not be computed conditionally. This choice is motivated by three reasons. First, when using precise free-form conditioning (such as LAION texts, which contain hashed product identifiers and product names) a symmetric encoding would enable a retrieval based solely on this information, completely disregarding the image query. Second, for discrete (categorical) conditioning it allows the presence of items of unknown category in the gallery, which is a situation that may occur in distractors. Third, these images only depict a single object, thus making referring information unnecessary. A similar setting is used by Asai et al. (2022).

Additionally, we provide a list of subsets sampled with replacement to be used for boostrapped estimation of confidence intervals on the metrics. We created 10 subsets of 1000 test products, and 10 subsets of 10K, 100K and 1M distractors. We also propose a validation set of 400 products with nearly 100K other distractors to monitor the training and for hyperparameter search.

## 5 EXPERIMENTS

We compare our method to various baselines on LRVS-F, using both category- and caption-based settings. We report implementation details before analyzing the results.

## 5.1 IMPLEMENTATION DETAILS

All our models take as input images of size $224 \times 224$, and output an embedding vector of 512 dimensions. We use CLIP weights as initialization, and then train our models for 30 epochs with AdamW (Loshchilov & Hutter, 2019) and a maximum learning rate of $10^{-5}$ determined by a learning rate range test (Smith, 2017). To avoid distorting pretrained features (Kumar et al., 2022), we start by only training the final projection and new input embeddings (conditioning and positional) for a single epoch, with a linear warm-up schedule. We then train all parameters for the rest of the epochs with a cosine schedule.

We pad the images to a square with white pixels, before resizing the largest side to 224 pixels. During training, we apply random horizontal flip, and random resized crops covering at least 80% of the image area. We evaluate the Recall at 1 (R@1) of the model on the validation set at each epoch, and report test metrics (recall and categorical accuracy) for the best performing validation checkpoint.

We used mixed precision and sharded loss to run our experiments on multiple GPUs. B/32 models were trained for 6 hours on 2 V100 GPUs, with a total batch size of 360. B/16 were trained for 9 hours on 12 V100, with a batch size of 420. Batch sizes were chosen to maximize GPU memory use.

## 5.2 RESULTS

**Detection-based Baseline**    We leveraged the recent Grounding DINO (Liu et al., 2023) and Segment Anything (Kirillov et al., 2023) to create a baseline approach based on object detection and segmentation. In this setting, we feed the model the query image and conditioning information, which can be either the name of the category or a caption. Subsequently, we use the output crops or masks to train a ViT following the aforementioned procedure. Please refer to Tab. 1 for the results.

Initial experiments conducted with pretrained CLIP features showed a slight preference toward segmenting the object. However, training the image encoder revealed that superior performances can be attained by training the network on crops. Our supposition is that segmentation errors lead to definitive loss of information, whereas the network's capacity is sufficient for it to learn to disregard irrelevant information and recover from a badly cropped image.

Overall, using Grounding DINO makes for a strong baseline. However, it is worth highlighting that the inherent imprecision of category names frequently results in overly large bounding boxes, which in turn limits the performances of the models. Indeed, adding more information into the dataset such as bounding boxes with precise categories would help, yet this would compromise the scalability of the model as such data is costly to obtain. Conversely, the more precise boxes produced by the caption-based model reach $67.8\%$R@1 against 2M distractors.

Table 1: Comparisons of results on LRVS-F for localization-based models. For 0, 10K, 100K and 1M distractors, we report bootstrapped means and standards deviations estimated from 10 randomly sampled sets. We observe superior performances from the caption-based models, due to the precision of the caption which leads to better detections.

| | | Distractors → | +10K | | +100K | | +1M | | +2M | |
|---|---|---|---|---|---|---|---|---|---|---|
| Condi. | Preprocessing | Embedding | %R@1 | %Cat@1 | %R@1 | %Cat@1 | %R@1 | %Cat@1 | %R@1 | %Cat@1 |
| **Category** | Gr. DINO-T + SAM-B | CLIP ViT-B/32 | 16.9 ±1.45 | 67.4 ±1.70 | 8.9 ±0.79 | 65.6 ±1.93 | 4.4 ±0.44 | 64.5 ±1.48 | 2.9 | 64.0 |
| | Gr DINO-T + SAM-B | ViT-B/32 | 83.0 ±1.06 | 94.6 ±0.75 | 69.4 ±1.36 | 92.0 ±0.67 | 53.1 ±1.63 | 90.0 ±0.77 | 46.4 | 89.2 |
| | Gr. DINO-T | ViT-B/32 | 88.7 ±0.74 | 96.4 ±0.55 | 77.0 ±1.79 | 94.3 ±0.82 | 62.8 ±1.92 | 92.2 ±1.26 | 56.0 | 91.8 |
| | Gr. DINO-B | ViT-B/16 | 89.9 ±0.87 | 96.2 ±0.77 | 80.8 ±1.35 | 94.5 ±0.73 | 68.8 ±2.17 | 93.2 ±0.90 | 62.9 | 92.5 |
| **Caption** | Gr. DINO-T + SAM-B | CLIP ViT-B/32 | 27.3 ±1.29 | 72.9 ±1.68 | 16.3 ±0.86 | 71.1 ±1.17 | 9.1 ±0.73 | 70.1 ±1.56 | 6.2 | 69.8 |
| | Gr. DINO-T + SAM-B | ViT-B/32 | 83.5 ±1.56 | 94.6 ±0.39 | 72.2 ±1.59 | 93.0 ±0.42 | 56.5 ±1.61 | 90.9 ±0.74 | 50.8 | 90.2 |
| | Gr. DINO-T | ViT-B/32 | 89.7 ±0.76 | 96.7 ±0.74 | 79.0 ±0.82 | 95.1 ±0.74 | 65.4 ±2.03 | 93.1 ±1.14 | 59.0 | 92.0 |
| | Gr. DINO-B | ViT-B/16 | 91.6 ±0.77 | 97.6 ±0.31 | 83.6 ±0.93 | 96.1 ±0.60 | 73.6 ±1.49 | 94.7 ±0.64 | 67.8 | 94.3 |

Table 2: Comparisons of results on LRVS-F for unconditional, category-based and caption-based models. For 0, 10K, 100K and 1M distractors, we report bootstrapped means and standards deviations from 10 randomly sampled sets. Our CondViT-B/16 outperforms other methods for both groups.

| | Distractors → | +10K | | +100K | | +1M | | +2M | |
|---|---|---|---|---|---|---|---|---|---|
| Model | | %R@1 | %Cat@1 | %R@1 | %Cat@1 | %R@1 | %Cat@1 | %R@1 | %Cat@1 |
| ViT-B/32 | | 85.6 ±1.08 | 93.7 ±0.31 | 73.4 ±1.35 | 90.9 ±0.78 | 58.5 ±1.37 | 87.8 ±0.86 | 51.7 | 86.9 |
| ViT-B/16 | | 88.4 ±0.88 | 94.8 ±0.52 | 79.0 ±1.02 | 92.3 ±0.73 | 66.1 ±1.21 | 90.2 ±0.92 | 59.4 | 88.8 |
| ASEN$_g$ (Dong et al., 2021) | | 63.1 ±1.50 | 76.3 ±1.26 | 46.1 ±1.21 | 68.5 ±0.84 | 29.8 ±1.86 | 62.9 ±1.27 | 24.1 | 62.0 |
| ViT-B/32 + Filt. | | 88.9 ±1.01 | — | 76.8 ±1.24 | — | 62.0 ±1.31 | — | 55.1 | — |
| CondViT-B/32 - Category | | 90.9 ±0.98 | 99.2 ±0.31 | 80.2 ±1.55 | 98.8 ±0.39 | 65.8 ±1.42 | 98.4 ±0.65 | 59.0 | 98.0 |
| ViT-B/16 + Filt. | | 90.9 ±0.88 | — | 81.9 ±0.87 | — | 68.9 ±1.11 | — | 62.4 | — |
| CondViT-B/16 - Category | | 93.3 ±1.04 | 99.5 ±0.25 | 85.6 ±1.06 | 99.2 ±0.35 | 74.2 ±1.82 | 99.0 ±0.42 | 68.4 | 98.8 |
| CoSMo (Lee et al., 2021) | | 88.3 ±1.30 | 97.6 ±0.45 | 76.1 ±1.85 | 96.0 ±0.32 | 59.1 ±1.42 | 94.7 ±0.40 | 52.1 | 94.8 |
| CLIP4CIR (Baldrati et al., 2023) | | 92.9 ±0.64 | 99.0 ±0.33 | 81.9 ±1.63 | 98.1 ±0.68 | 66.9 ±2.05 | 96.5 ±0.67 | 59.1 | 95.5 |
| CondViT-B/32 - Caption | | 92.7 ±0.77 | 99.1 ±0.30 | 82.8 ±1.22 | 98.7 ±0.40 | 68.4 ±1.50 | 98.1 ±0.43 | 62.1 | 98.0 |
| CondViT-B/16 - Caption | | 94.2 ±0.90 | 99.4 ±0.37 | 86.4 ±1.13 | 98.9 ±0.49 | 74.6 ±1.65 | 98.4 ±0.58 | 69.3 | 98.2 |

**Categorical Conditioning**    We compare our method with categorical detection-based approaches, and unconditional ViTs finetuned on our dataset. To account for the extra conditioning information used in our method, we evaluated the latter on filtered indexes, with only products belonging to the

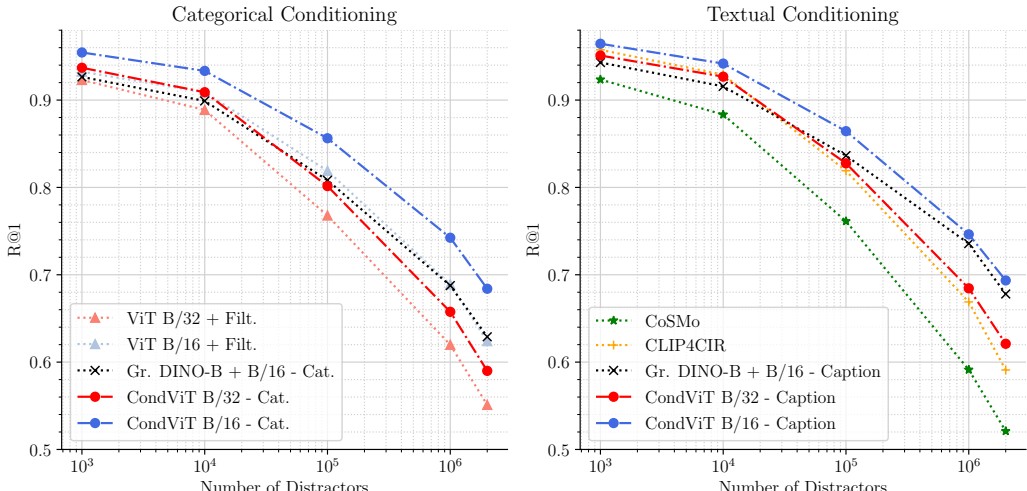

Figure 4: R@1 with repects to number of added distractors, evaluated on the entire test set. Please refer to Tab. 1 and 2 for bootstrapped metrics and confidence intervals. Our categorical CondViT-B/16 reaches the performances of the best caption-based models, while using a sparser conditioning.

correct category. We did not try to predict the item of interest from the input picture, and instead consider it as a part of the query. We also report unfiltered metrics for reference. Results are in Tab. 2.

Training the ViTs on our dataset greatly improves their performances, both in terms of R@1 and categorical accuracy. Filtering the gallery brings a modest mean gain of $2 - 4\%$R@1 across all quantities of distractors (Fig. 4b), reaching $62.4\%$R@1 for 2M distractors with a ViT-B/16 architecture. In practice, this approach is impractical as it necessitates computing and storing an index for each category to guarantee a consistent quantity of retrieved items. Moreover, a qualitative evaluation of the filtered results reveals undesirable behaviors. When filtering on a category divergent from the network's intrinsic focus, we observe the results displaying colors and textures associated with the automatically focused object rather than the requested one.

We also compare with ASEN (Dong et al., 2021) trained on our dataset using the authors' released code. This conditional architecture uses a global and a local branch with conditional spatial attention modules, respectively based on ResNet50 and ResNet34 backbones, with explicit ROI cropping. However in our experiments the performances decrease with the addition of the local branch in the second training stage, even after tuning the hyperparameters. We report results for the global branch.

We train our CondViT using the categories provided in our dataset, learning an embedding vector for each of the 10 clothing categories. For the $i$-th product in the batch, we randomly select in the associated data a simple image $x_s$ and its category $c_s$, and a complex image $x_c$. We then compute their embeddings $z_i^A = \phi(x_c, c_s), z_i^B = \phi(x_s)$. We also experimented with symmetric conditioning, using a learned token for the gallery side (see Appendix B.3).

Our categorical CondViT-B/16, with $68.4\%$R@1 against 2M distractors significantly outperforms all other category-based approaches (see Fig. 4, left) and maintains a higher categorical accuracy. Furthermore, it performs similarly to the detection-based method conditioned on richer captions, while requiring easy-to-aquire coarse categories. It does so without making any assumption on the semantic nature of these categories, and adding only a few embedding weights (7.7K parameters) to the network, against 233M parameters for Grounding DINO-B. We confirm in Appendix B.2 that its attention is localized on different objects depending on the conditioning.

**Textual Conditioning**  To further validate our approach, we replaced the categorical conditioning with referring expressions, using our generated BLIP2 captions embedded by a Sentence T5-XL model (Ni et al., 2022). We chose this model because it embeds the sentences in a 768-dimensional vector, allowing us to simply replace the categorical token. We pre-computed the caption embeddings, and randomly used one of them instead of the product category at training time. At test time, we used the first caption.

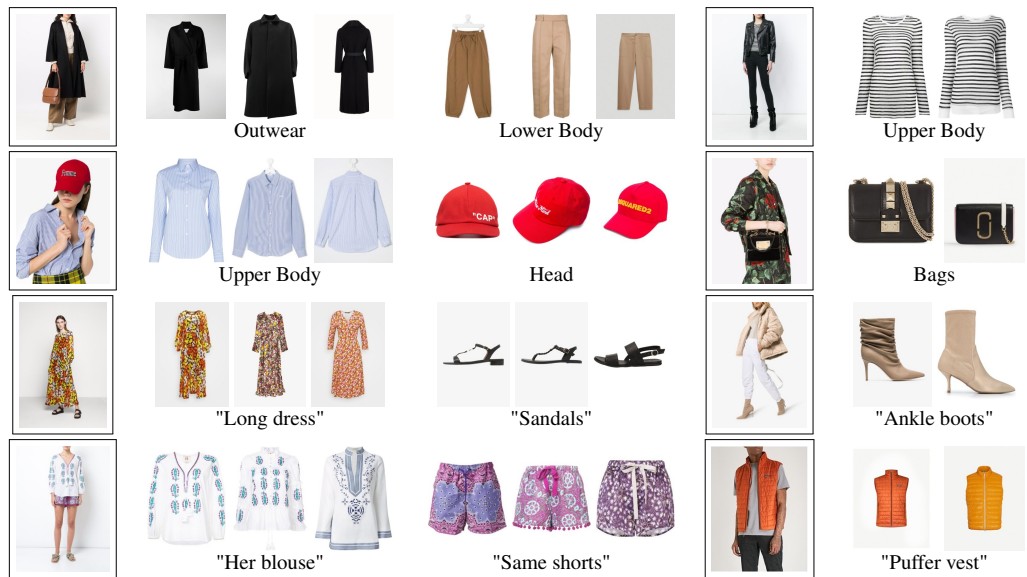

Figure 5: Qualitative results for our categorical (first 2 rows) and textual (last 2 rows) CondViT-B/16. We use free-form textual queries instead of BLIP2 captions to illustrate realistic user behavior, and retrieve from the whole test gallery. See Fig. 8 and 9 in the Appendix for more qualitative results.

In Tab. 2, we observe a gain of 3.1%R@1 for the CondViT-B/32 architecture, and 0.9%R@1 for CondViT-B/16, compared to categorical conditioning against 2M distractors, most likely due to the additional details in the conditioning sentences. When faced with users, this method allows for more natural querying, with free-form referring expressions. See Figure 5 for qualitative results.

We compare these models with CIR methods: CoSMo (Lee et al., 2021) and CLIP4CIR (Baldrati et al., 2023). Both use a compositor network to fuse features extracted from the image and accompanying text. CoSMo reaches performances similar to an unconditional ViT-B/32, while CLIP4CIR performs similarly to our textual CondViT-B/32. We hypothesize that for our conditional feature extraction task, early conditioning is more effective than modifying embeddings through a compositor at the network's end. Our CondViT-B/16 model significantly outperforms all other models and achieves results comparable to our caption-based approach using Grounding DINO-B (see Fig. 4, right). As the RVS task differs from CIR, despite both utilizing identical inputs, this was anticipated. Importantly, CondViT-B/16 accomplishes this without the need for explicit detection steps or dataset-specific preprocessing. Notably, we observe that our models achieve a categorical accuracy of 98% against 2M distractors, surpassing the accuracy of the best corresponding detection-based model, which stands at 94.3%.

## 6 CONCLUSION & LIMITATIONS

We studied an approach to image similarity in fashion called Referred Visual Search (RVS), which introduces two significant contributions. Firstly, we introduced the LAION-RVS-Fashion dataset, comprising 272K fashion products and 842K images. Secondly, we proposed a simple weakly-supervised learning method for extracting referred embeddings. Our approach outperforms strong detection-based baselines. These contributions offer valuable resources and techniques for advancing image retrieval systems in the fashion industry and beyond.

However, one limitation of our approach is that modifying the text description to refer to something not present or not easily identifiable in the image does not work effectively. For instance, if the image shows a person carrying a green handbag, a refined search with "red handbag" as a condition would only retrieve a green handbag. The system may also ignore the conditioning if the desired item is small or absent in the database. Examples of such failures are illustrated in Appendix B.4. Additionally, extending the approach to more verticals would be relevant.

## 7 ETHICS STATEMENT

**Harmful and Private Content.** Our dataset is a subset of the publicly released LAION 5B dataset, enriched with synthetic metadatas (categories, captions, product identifiers). However, our process began by curating a subset of domains, focusing exclusively on domains affiliated with well-known fashion retailers and URLs containing product identifiers. As such, these images come from large commercial fashion catalogs. Our dataset contains images that appear in online fashion catalogs and does not contain harmful or disturbing images. Most of the images are pictures of isolated attire on neutral backgrounds. Images depicting people are all extracted from professional photoshoots, with all the ethical and legal considerations that are common practices in the fashion retail industry.

We release our dataset only for research purposes as a benchmark to study Referred Visual Search where no public data exists, which is a problem for reproducibility. This is an object-centric instance retrieval task that aims to control more precisely the content of image embeddings. On this dataset, to optimize the performances, embeddings should only contain information regarding the referred garment, rather than the model wearing it.

**Dataset Biases.** Our dataset lacks metadata for a comprehensive exploration of bias across gender and ethnicity. However, based on an inspection of a random sample of 1000 images, we estimate that roughly 2/3 of the individuals manifest discernible feminine physical attributes or attire.

Among the cohort of 22 fashion retailers featured in our dataset, 14 are from the European Union, 7 are from the United States, and the remaining one is from Russia. Thereby, even though these retailers export and sell clothing across the world, our dataset reproduces the biases of European and American fashion industries with respect to models' ethnicity and gender.

**Retrieval Systems.** The broader impact of this research is similar to other works on instance retrieval, with the significant advantage of producing embeddings that conceal information about personal identity and other undesired elements thanks to the conditioning.

## 8 REPRODUCIBILITY STATEMENT

The code used to produce the results will be released. Additionally, our training setup is described in Section 5.1.

The dataset will be publicly released. The method we used to create it is thoroughly described in Appendix A.2.

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

# APPENDIX

## A   DATASET

### A.1   SAMPLES

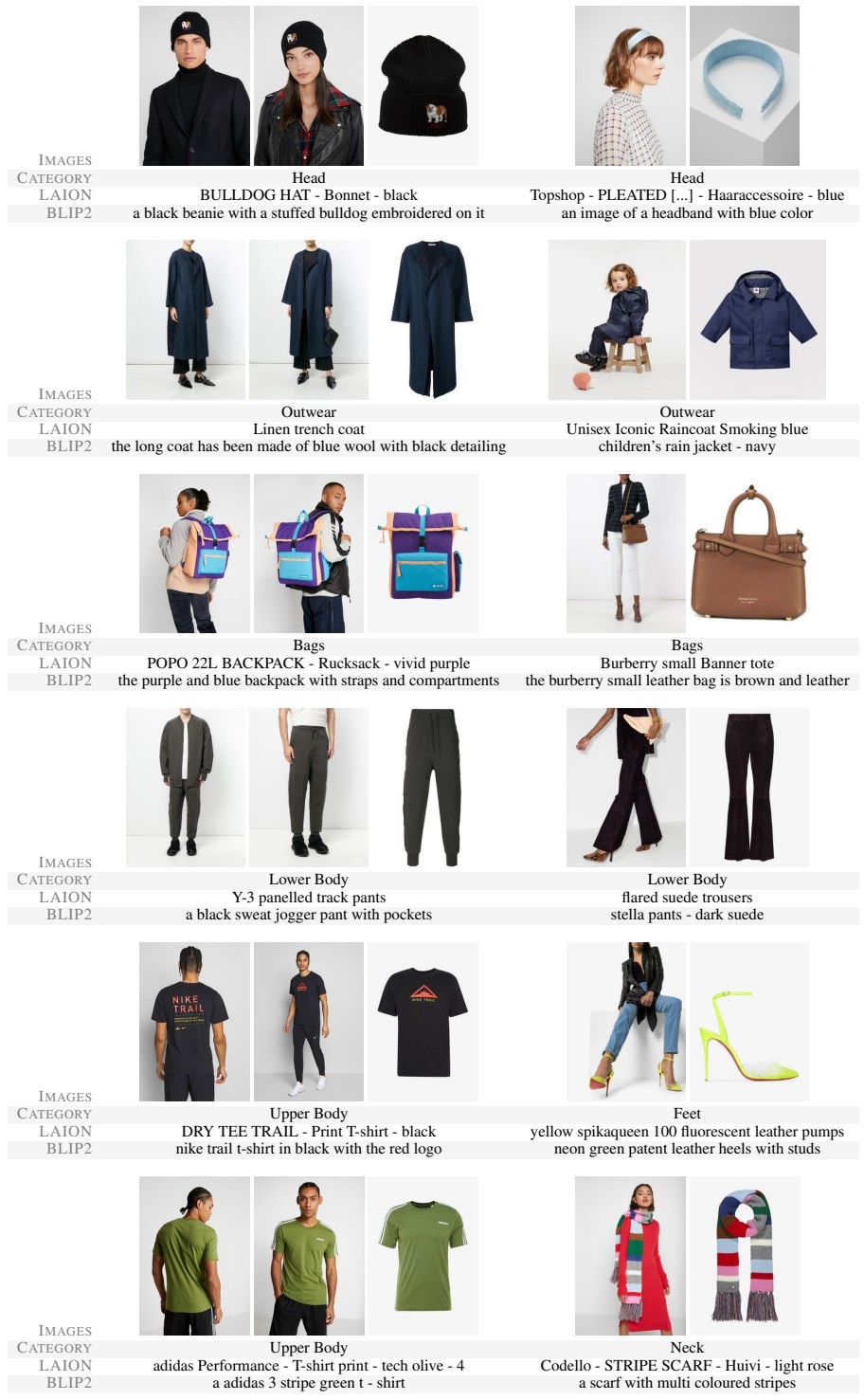

Figure 6: Additional samples from LRVS-F.

## A.2 CONSTRUCTION

**Image Collection:**   The raw data of LRVS-F are collected from a list of fashion brands and retailers whose content delivery network domains were found in LAION 5B. We used the automatically translated versions of LAION 2B MULTI and LAION 1B NOLANG to get english captions for all the products. This represents around 8M initial images.

We analyzed the format of the URLs for each domain, and extracted image and product identifiers using regular expressions when possible. We removed duplicates at this step using these identifiers, and put aside images without clear identifiers to be filtered and used as distractors later.

**Image Annotation:**   The additional metadata that we provide were generated using deep learning models. We generated indicators of the image complexity, classified the products in 11 categories, and added new image captions.

First, we used a model to classify the complexity of the images, trained with active learning. We started by automatically labeling a pool of images using information found in the URLs, before manually filtering the initial data, and splitting between training and validation. Then, we computed and stored the pre-projection representations extracted by OpenCLIP B16 for each image, and trained a 2-layers MLP to predict the category. After training, we randomly sampled 1000 unlabeled images and annotated the 100 with the highest prediction entropy, before splitting them between training and validation data. We repeated these 2 steps until reaching over 99% accuracy and labeled the entire dataset using this model.

We used a second model to automatically assign categories to the simple images. LAION captions are noisy, so instead of using them we used BLIP2 FlanT5-XL (Li et al., 2023) to answer the question "In one word, what is this object?". We gathered all the nouns from the answers, using POS tagging when the generated answer was longer, and grouped them in 11 categories (10 for clothing, 1 for non-clothing). We automatically created an initial pool of labeled data, which we manually filtered, before applying the same active learning process as above. We then annotated all the simple images with this model. Please refer to Appendix A.3 for the list of categories and their composition.

Finally, we automatically added new descriptions to the simple images, because the quality of some LAION texts was low. For example, we found partially translated sentences, or product identifiers. We generated 10 captions for each image using BLIP2 FlanT5-XL with nucleus sampling, and kept the two with largest CLIP similarity.

**Dataset Split:**   We grouped together images associated to the same product identifier and dropped the groups that did not have at least a simple and a complex image. We manually selected 400 of them for the validation set, and 2,000 for the test set. The distractors are all the images downloaded previously that were labeled as "simple" but not used in product groups. This mostly includes images for which it was impossible to extract any product identifier.

Finally, we used Locally Sensitive Hashing (LSH) with perceptual hash, and OpenCLIP B16 embeddings to remove duplicates. We created FAISS indexes based respectively on hamming distance and cosine similarity, automatically removing samples with extremely high similarity. We manually inspected samples near the threshold. We used this process on complex images from the training set to remove products duplicates, on train and test sets to reduce evaluation bias, and on gallery images and distractors for both the validation and test sets.

## A.3 COMPOSITION

We classified LRVS-F products into 11 distinct categories. Among these categories, 10 are specifically related to clothing items, which are organized based on their approximate location on the body. Additionally, there is one non-clothing category included to describe some distractors. Tab. 3 provides information regarding the counts of products within each category, as well as the data split. For a more detailed understanding of the clothing categories, Tab. 4 presents examples of fine-grained clothing items that are typically associated with each category.

Each product in our dataset is associated with at least one simple image and one complex image. In Figure 7, we depict the distribution of simple and complex images for each product. Remarkably,

we observe that the majority of products, accounting for 90% of the dataset, possess a single simple image and up to four complex images.

Table 3: Count of simple images (isolated items) across the dataset splits. Some training products are depicted in multiple simple images, hence the total higher than the number of unique identities.

| | Upper Body | Lower Body | Whole Body | Outwear | Bags | Feet | Neck | Head | Hands | Waist | NonClothing | Total |
|---|---|---|---|---|---|---|---|---|---|---|---|---|
| **Train** | 92 410 | 75 485 | 48 446 | 45 867 | 26 062 | 4 224 | 3 217 | 1 100 | 190 | 184 | - | 297 185 |
| **Val** | 80 | 80 | 80 | 80 | 60 | 6 | 6 | 4 | 2 | 2 | - | 400 |
| **Test** | 400 | 400 | 400 | 400 | 300 | 30 | 30 | 20 | 10 | 10 | - | 2 000 |
| **Val. Dist.** | 19 582 | 13 488 | 8 645 | 6 833 | 10 274 | 22 321 | 2 470 | 6 003 | 2 866 | 1 016 | 6 043 | 99 541 |
| **Test Dist.** | 395 806 | 272 718 | 172 385 | 136 062 | 203 390 | 448 703 | 50 881 | 121 094 | 57 271 | 19 853 | 121 851 | 2 000 014 |

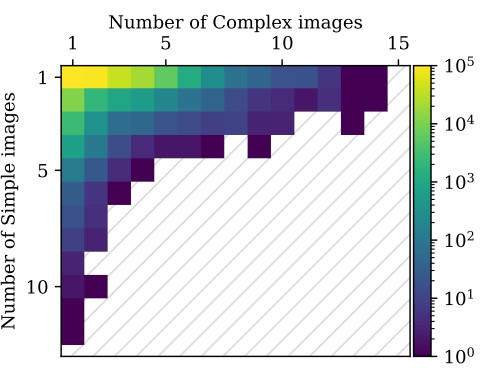

Table 4: Examples of sub-categories.

| CATEGORY | COMPOSITION |
|---|---|
| **Upper Body** | T-shirts, Shirts, Crop Tops, Jumper, Sweater … |
| **Lower Body** | Shorts, Pants, Leggings, Skirts … |
| **Whole Body** | Dress, Gown, Suits, Rompers … |
| **Outwear** | Coat, Jacket … |
| **Bags** | Handbags, Backpack, Luggage … |
| **Feet** | Shoes, Boots, Socks … |
| **Neck** | Scarves, Necklace … |
| **Head** | Hat, Cap, Glasses, Sunglasses, Earrings … |
| **Hands** | Gloves, Rings, Wristbands… |
| **Waist** | Belts |

Figure 7: Distribution of Simple and Complex images across products. 90% of the products have 1 simple image and up to 4 complex images.

# B  MODEL

## B.1  RETRIEVAL EXAMPLES

In this section, we show additional results for our categorical CondViT-B/16 and its textual variant trained with BLIP2 (Li et al., 2023) captions. We use test query images and the full test gallery with 2M distractors for the retrieval. Each query in the test set is exclusively associated with a single item. However, it should be noted that the we do not necessarily query for this item, so the queried product might not be in the gallery. Nevertheless, owing to the presence of 2M distractors, most queries can retrieve multiple viable candidates.

Fig. 8 shows that our categorical CondViT is able to extract relevant features across a wide range of clothing items, and propose a coherent retrieval especially for the main categories. There is still room for improvement on images depicting rare training categories like *Waist*, *Hands*, *Head* or *Neck*, and rare poses.

Fig. 9 presents improvements brought by textual conditioning captions over categorical conditioning. Using text embeddings allows for more natural querying, thanks to the robustness of our model to irrelevant words. However, this robustness comes at the cost of ignoring appearance modifications.

## B.2  ATTENTION MAPS

We propose a visualization of the attention maps of our ViT-B/16, ASEN, and our categorical CondViT-B/16 in Fig. 10. We compare attention in the last layer of the transformers with the Spatial Attention applied at the end of ASEN's global branch. We observe that the attention mechanism in the transformers exhibits a notably sparse nature, selectively emphasizing specific objects within the input scene. Conversely, ASEN demonstrates a comparatively less focused attention distribution.

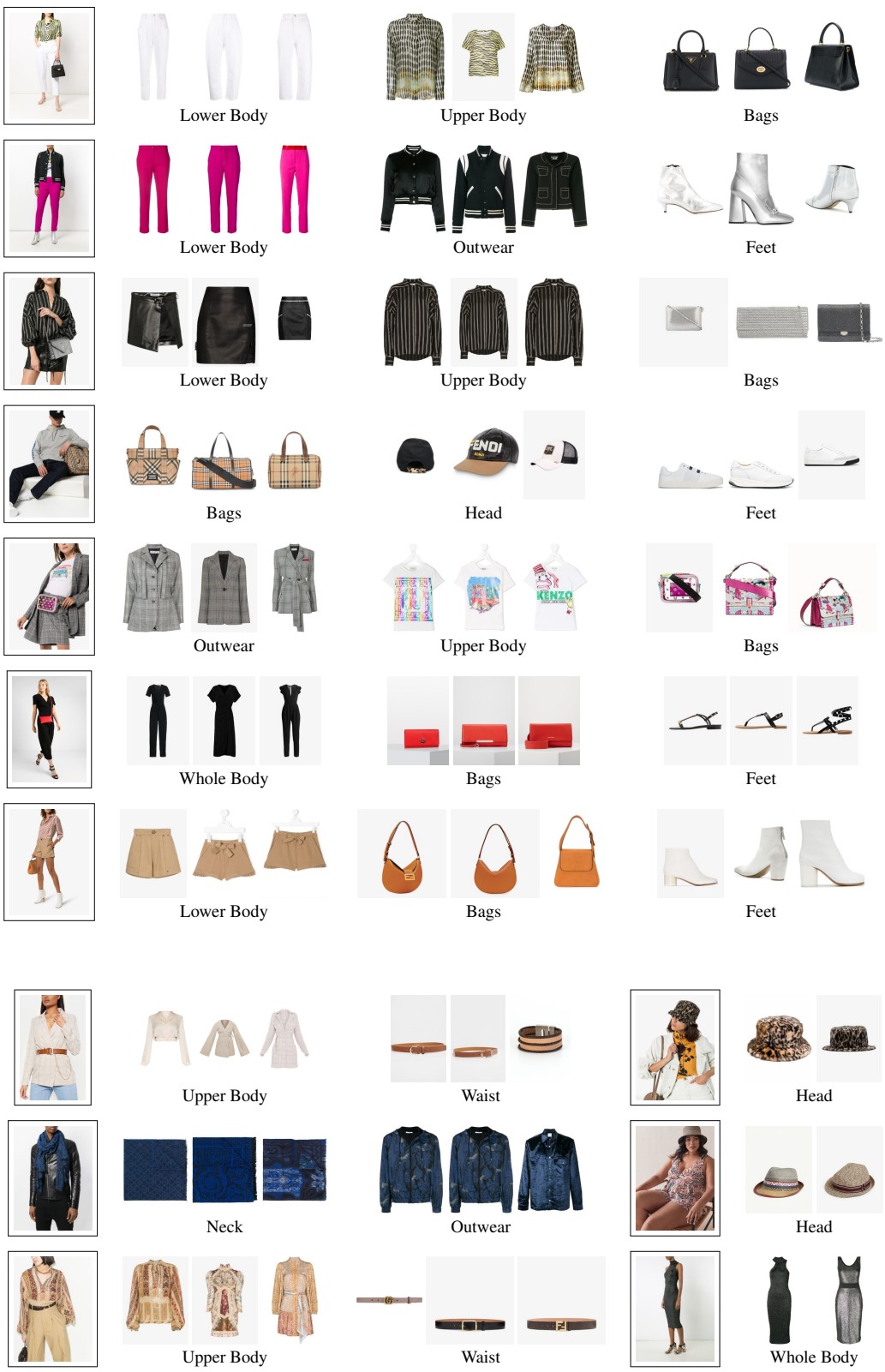

Figure 8: Qualitative results of our Conditional ViT-B/16 on LRVS-F test set.

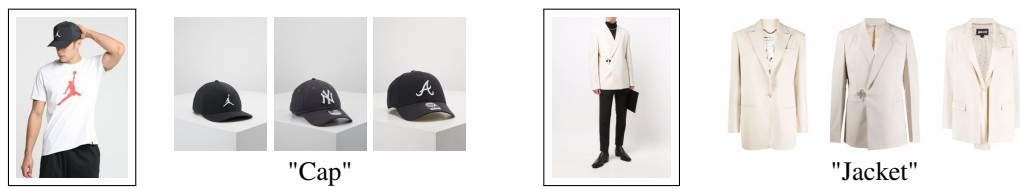

(a) Top-3 retrieval for normal user queries. Even though the BLIP2 captions were more detailed, using a single word as a query produces the expected result.

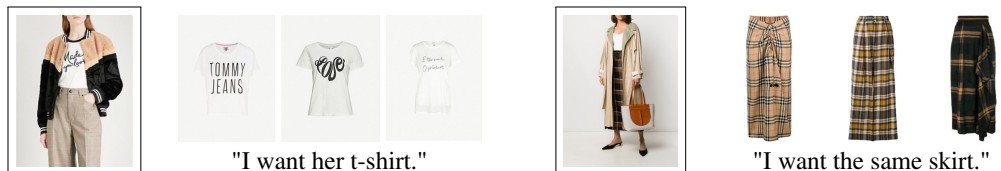

(b) Top-3 retrieval for noisy user queries. Our model is robust to expression of user intent and can focus on the designated object.

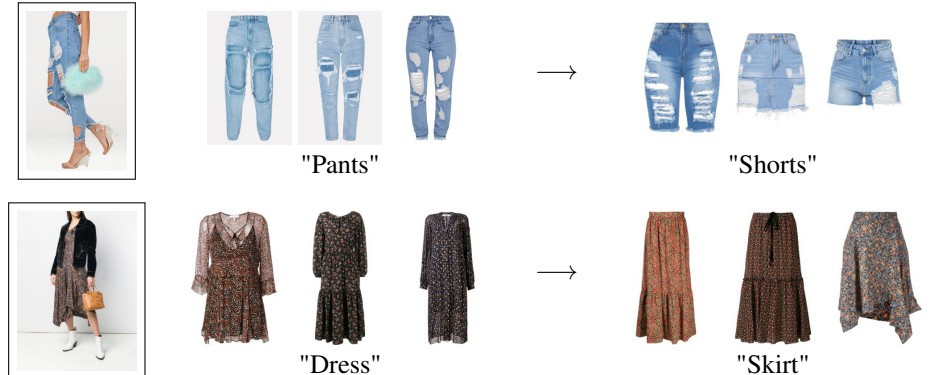

(c) Top-3 retrieval for queries with item modifications. In some circumstances, a textual query can influence the result to slightly modify the type of retrieved items, *e.g.* exchanging shorts and pants or skirts and dresses.

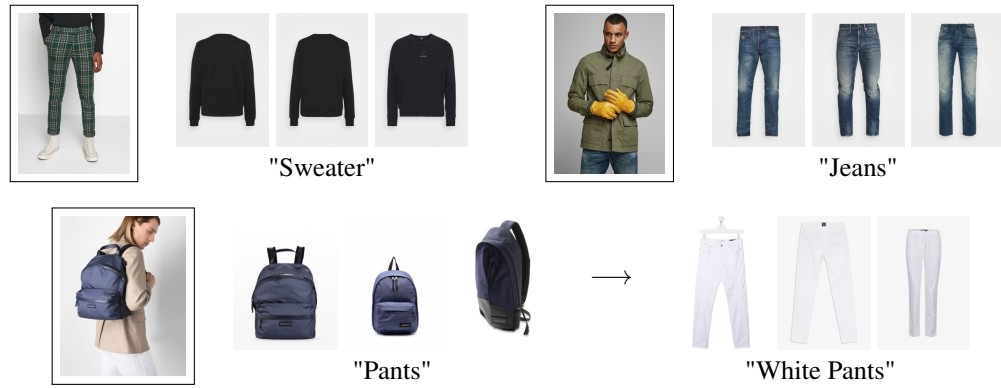

(d) Top-3 retrieval for out-of-frame items. If the network fails, we find that precising the query can help.

Figure 9: Retrieved items for queries in LRVS-F test set with our textual CondViT-B/16. (a) shows results for normal, concise use. (b) shows results with more verbose queries. (c) shows queries influencing the type of results. (d) show results for out-of-frame items.

Surprisingly, the unconditional ViT model exhibits a strong focus on a single object of the scene, while the attention of our CondViT dynamically adjusts in response to the conditioning information.

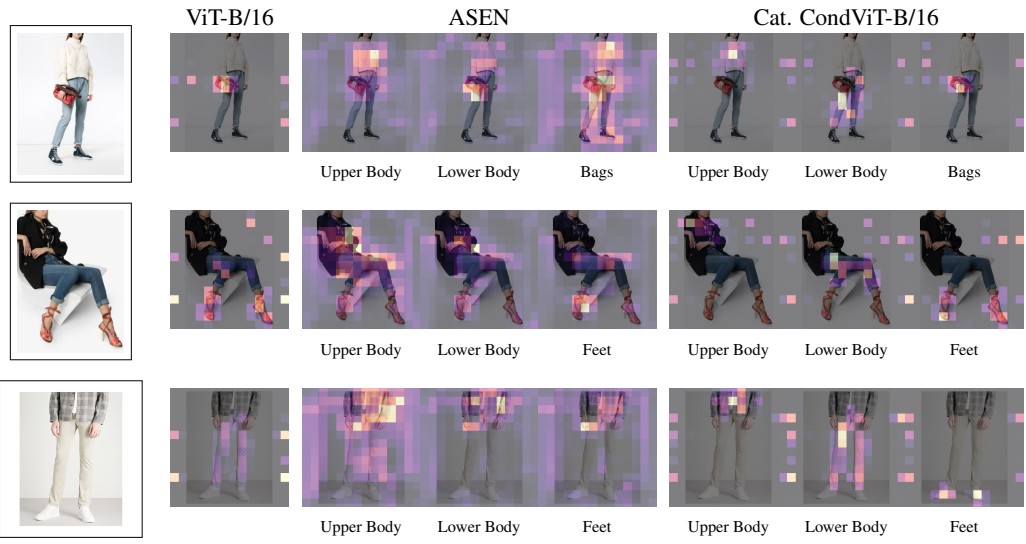

Figure 10: Attention maps. For ViT-B/16 and CondViT-B/16, we display the maximum attention from the CLS token to the image tokens across all heads in the last layer, and observe sparse maps. For ASEN, we display the attention returned by the Spatial Attention module of the global branch, and observe more diffuse maps. All maps are normalized to [0-1].

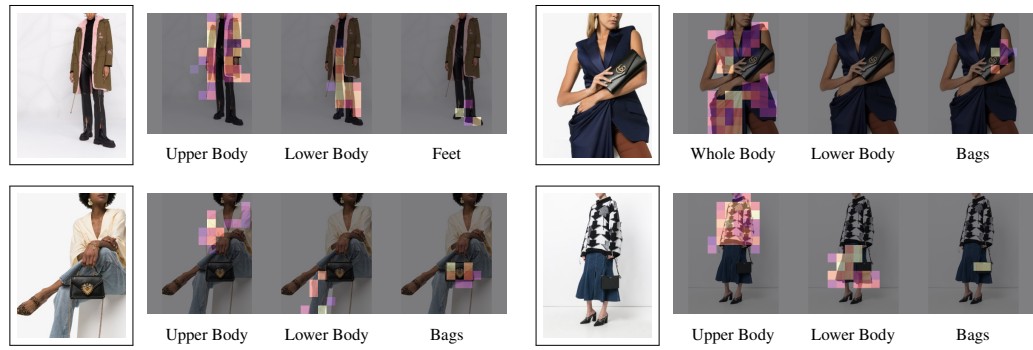

Figure 11: Visualization of the thresholded first component of image tokens in our CondViT-B/16. This component enables separation of the background, foreground, and focused object.

Figure 11 shows the patch features extracted by our models with principal component analysis (PCA) computed on all image tokens in the last layer of our CondViT-B/16 model across the test queries. Similarly to Oquab et al. (2023), we find that applying a threshold on the first component enables effective separation of the background from the foreground. Intriguingly, we observe that employing a higher threshold not only accomplishes the aforementioned separation but also yields cleaner visualizations by isolating the conditionally selected object. We also observe instances where the network encounters difficulties in detecting the referenced object, resulting in a notable absence of tokens surpassing the established threshold.

## B.3 ABLATION STUDIES

**Insertion Depth.** We study the impact of the insertion depth of our additional conditioning token by training a series of CondViT-B/32, concatenating the conditioning token before different encoder blocks for each one of them.

Fig. 12 indicates that early concatenation of the conditioning token is preferable, as we observed a decrease in recall for deep insertion (specifically, layers 10-12). However, there was no statistically significant difference in performance between layers 1-8. Consequently, we decided to concatenate the token at the very beginning of the model. We hypothesize that the presence of residual connections in our network enables it to disregard the conditioning token until it reaches the optimal layer. The choice of this layer may depend on factors such as the size of the ViT model and the characteristics of the dataset being used.

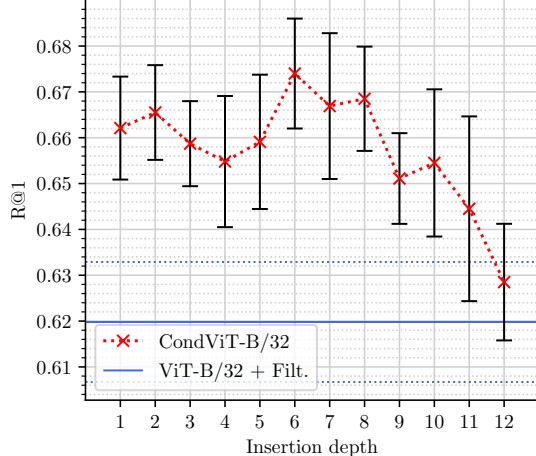

Figure 12: R@1 on the test set with respect to the insertion depth of the conditioning token. Error bars represent the bootstrapped estimation of the standard deviation across 10 splits. Late insertion degrades performance, but no significant difference can be seen among the first layers.

**Asymetric Conditioning.** We experiment with using conditioning for the simple images too, using a single learned "empty" token for all the simple images. We denote this token $c_\emptyset$. Then for each simple image $x_s$ we compute its embedding as $\phi(x_s, c_\emptyset)$.

Results in Tab. 5 show that there is no really significant difference between both approaches, even though CondViT-B/16 results are better without this additional token for large amounts of distractors ($\geq$ 100K). We choose to keep an asymmetric embedding process.

Table 5: Comparison of symmetric and asymmetric conditioning on LRVS-F test set. We report bootstrapped mean and standard deviation on the test set. There is no significant difference between the configurations. Bold results indicate a difference of more than 1%.

| Distractors → | +0 | | +10K | | +100K | | +1M | | +2M | |
|---|---|---|---|---|---|---|---|---|---|---|
| Model | %R@1 | %Cat@1 | %R@1 | %Cat@1 | %R@1 | %Cat@1 | %R@1 | %Cat@1 | %R@1 | %Cat@1 |
| CondViT-B/32 | 97.0 ±0.57 | 100 ±0.07 | 90.9 ±0.98 | 99.2 ±0.31 | 80.2 ±1.55 | 98.8 ±0.39 | 65.8 ±1.42 | 98.4 ±0.65 | 59.0 | 98.0 |
| CondViT-B/32 + $c_\emptyset$ | 96.8 ±0.94 | 100 ±0.10 | 91.1 ±1.04 | 99.3 ±0.24 | 79.9 ±1.35 | 99.0 ±0.21 | 66.0 ±1.36 | 98.3 ±0.46 | 59.6 | 98.2 |
| CondViT-B/16 | 97.7 ±0.21 | 99.8 ±0.12 | 93.3 ±1.04 | 99.5 ±0.25 | **85.6** ±1.06 | 99.2 ±0.35 | **74.2** ±1.82 | 99.0 ±0.42 | **68.4** | 98.8 |
| CondViT-B/16 + $c_\emptyset$ | 97.8 ±0.32 | 99.9 ±0.11 | 93.2 ±0.79 | 99.5 ±0.16 | 84.4 ±1.16 | 99.0 ±0.29 | 72.5 ±1.88 | 98.8 ±0.42 | 66.5 | 98.0 |

### B.4 TEXTUAL CONDITIONING — FAILURE CASES

We finally present limitations of our textual CondViT-B/16 in Fig. 13. Firstly, when faced with failure in identifying the referenced object, our model resorts to selecting the salient object instead. Additionally, our model ignores queries with color or texture modifications, returning objects as depicted in the query image.

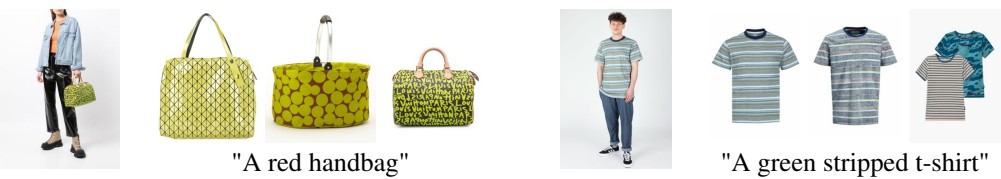

(a) Top-3 retrieval for queries trying to modify color of an item. We find such modifications to be mostly ignored by the model.

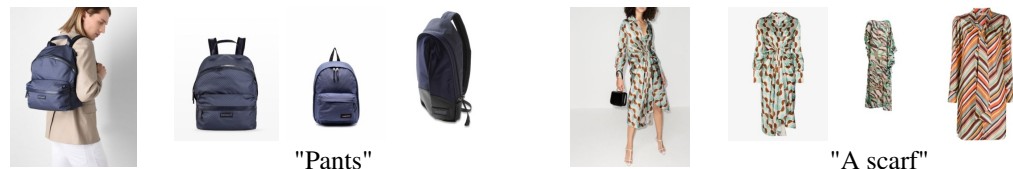

(b) Top-3 retrieval for missed queries. For hard queries, or queries about an item not represented in the picture we find a tendency to default to the salient item in the image.

Figure 13: Retrieved items showing failure cases of our textual CondViT-B/16. (a) shows that the network disregards color clues. (b) shows that the network defaults to the salient item when the query is too hard or not represented.

## C    DATASHEET

### C.1    MOTIVATIONS

Q1. **For what purpose was the dataset created?** *Was there a specific task in mind? Was there a specific gap that needed to be filled? Please provide a description.*

This dataset has been created to provide public training data and a benchmark for the Referred Visual Search (RVS) task, for research purposes. The task is new, and thereby no other dataset existed to tackle it.

Q2. **Who created the dataset (e.g., which team, research group) and on behalf of which entity (e.g., company, institution, organization)?**

[Anonymized].

Q3. **Who funded the creation of the dataset?** *If there is an associated grant, please provide the name of the grantor and the grant name and number.*

[Anonymized].

Q4. **Any other comments ?**

No.

### C.2    COMPOSITION

Q5. **What do the instances that comprise the dataset represent (e.g., documents, photos, people, countries)?** *Are there multiple types of instances (e.g., movies, users, and ratings; people and interactions between them; nodes and edges)? Please provide a description.*

Instances of this dataset are URLs from online catalogs of fashion retailers, and a such they depict either fashion products or models wearing them.

Q6. **How many instances are there in total (of each type, if appropriate)?**

In total, there are :

- 299,585 target simple images
- 486,995 complex images
- 59,938 partial complex images
- 2,099,555 additional simple images, not linked to any product, that serve as distractors.

Q7. **Does the dataset contain all possible instances or is it a sample (not necessarily random) of instances from a larger set?** *If the dataset is a sample, then what is the larger set? Is the sample representative of the larger set (e.g., geographic coverage)? If so, please describe how this representativeness was validated/verified. If it is not representative of the larger set, please describe why not (e.g., to cover a more diverse range of instances, because instances were withheld or unavailable).*

Our dataset is merely a small fashion subset of LAION-5B, which is itself a subset of the CommonCrawl dataset. We only selected a small amount of retailers and brands, mostly with European and American influence. As such, it is only a sample of fashion images, and is not representative of retailers and brands from other geographical areas.

Q8. **What data does each instance consist of? "Raw" data (e.g., unprocessed text or images) or features?** *In either case, please provide a description.*

Instances of the dataset are URLs of images, accompanied by various metadatas. Among them, their widths, heights, probabilities of containing a watermark, probabilities of being NSFW, associated texts (translated to english when needed) and original languages all originate from the LAION-5B dataset, and we refer the reader to this dataset for additional information. They are not used in the benchmark but we report them for ease of use and safety.

We added mutiple synthetic labels to the images. First, a type, COMPLEX when the image depicts a scene, with a model, SIMPLE when it is an isolated product. There also exist a PARTIAL COMPLEX category, for scene images that are zoomed-in and do not contain the entire product. Second, a product identifier, allowing to group images depicting the same product. Each simple target image is further described by a category, following the taxonomy described in this paper, and 2 BLIP2-FlanT5XL captions.

Q9. **Is there a label or target associated with each instance?** *If so, please provide a description.*

We added categories and captions associated with each simple training image, but they are intended to be used as inputs to the models. The product identifier could be seen as a target as we propose a product retrieval task.

Q10. **Is any information missing from individual instances?** *If so, please provide a description, explaining why this information is missing (e.g., because it was unavailable). This does not include intentionally removed information, but might include, e.g., redacted text.*

Yes, complex images often depict multiple objects, but are linked with only one product in this dataset. They are registered in fashion catalogs with the intent to showcase a specific product, and as such we were not able to extract more information.

Q11. **Are relationships between individual instances made explicit (e.g., users' movie ratings, social network links)?** *If so, please describe how these relationships are made explicit.*

Yes, we provide a synthetic product identifier for each image, allowing to group simple and complex images depicting the same product.

Q12. **Are there recommended data splits (e.g., training, development/validation, testing)?** *If so, please provide a description of these splits, explaining the rationale behind them.*

Yes. We selected 400 products and 99,541 distractors to create a validation set. We also selected 2,000 products and 2,000,014 distractors to create a large test set. We selected the products so that their category distribution roughly match their distribution in the training set.

Q13. **Are there any errors, sources of noise, or redundancies in the dataset?** *If so, please provide a description.*

We created most of the new labels synthetically, using classifiers and captioners, so they contain some noise. However, by randomly sampling images and manually verifying their labels, we find an empiric error rate of 1/1000 for training complex images, 0/1000 for training simple images, and 3/1000 for distractors. Regarding the categories, we find an empiric error rate of less than 1%, with the confusions mostly stemming from semantically

similar categories and images where object scale was ambiguous in isolated settings (long shirt against short dress, wristband against hairband).

The BLIP2 captions that we provide are of good quality and increase the mean CLIP similarity with the image of +7.4%. However, as synthetic captions, they are not perfect and sometimes contain hallucinations.

There are some redundancies in the distractors sets.

Q14. **Is the dataset self-contained, or does it link to or otherwise rely on external resources (e.g., websites, tweets, other datasets)?** *If it links to or relies on external resources, a) are there guarantees that they will exist, and remain constant, over time; b) are there official archival versions of the complete dataset (i.e., including the external resources as they existed at the time the dataset was created); c) are there any restrictions (e.g., licenses, fees) associated with any of the external resources that might apply to a future user? Please provide descriptions of all external resources and any restrictions associated with them, as well as links or other access points, as appropriate.*

No, the dataset relies on external links to the World Wide Web. We are unable to offer any guarantees of the existence of the images over time. We do not own the rights of these images, and as such do not provide any archival version of the complete dataset. These copyrights might contains restriction about the images use. We encourage any user of the dataset to inquire about these copyrights.

Q15. **Does the dataset contain data that might be considered confidential (e.g., data that is protected by legal privilege or by doctor–patient confidentiality, data that includes the content of individuals' non-public communications)?** *If so, please provide a description.*

No. This dataset only contains samples from online fashion catalogs, and as such does not contain any confidential or personal data.

Q16. **Does the dataset contain data that, if viewed directly, might be offensive, insulting, threatening, or might otherwise cause anxiety?** *If so, please describe why.*

No. This dataset contains samples from online fashion catalogs, that result from professional photoshoots with the objective to be as appealing a possible to a large amount of customers.

Q17. **Does the dataset relate to people?** *If not, you may skip the remaining questions in this section.*

Models are present in the complex images. However, the sole focus of our dataset is the fashion items they are wearing, and most of the images are isolated objects. It does not contain any private or personal information.

Q18. **Does the dataset identify any subpopulations (e.g., by age, gender)?** *If so, please describe how these subpopulations are identified and provide a description of their respective distributions within the dataset.*

No, the dataset does not contain any metadata allowing to identify any subpopulation.

Q19. **Is it possible to identify individuals (i.e., one or more natural persons), either directly or indirectly (i.e., in combination with other data) from the dataset?** *If so, please describe how.*

It might be possible to identify models using face recognition, but it would require external data.

Q20. **Does the dataset contain data that might be considered sensitive in any way (e.g., data that reveals racial or ethnic origins, sexual orientations, religious beliefs, political opinions or union memberships, or locations; financial or health data; biometric or genetic data; forms of government identification, such as social security numbers; criminal history)?** *If so, please provide a description.*

No.

Q21. **Any other comments ?**

No.

### C.3 COLLECTION PROCESS

Q22. **How was the data associated with each instance acquired?** *Was the data directly observable (e.g., raw text, movie ratings), reported by subjects (e.g., survey responses), or indirectly inferred/derived from other data (e.g., part-of-speech tags, model-based guesses for age or language)? If data was reported by subjects or indirectly inferred/derived from other data, was the data validated/verified? If so, please describe how.*

The initial data was acquired from LAION-5B, a subset of CommonCrawl. Please refer to their work for details about this initial data acquisition. The additional labels were synthetically generated by deep neural networks, based on manually annotated data, and a pretrained captioner.

Q23. **What mechanisms or procedures were used to collect the data (e.g., hardware apparatus or sensor, manual human curation, software program, software API)?** *How were these mechanisms or procedures validated?*

We manually curated domains and manually designed regular expressions to extract product identifiers from the URLs. The additional labels and captions are synthetic. We validated the quality of the labels by mesuring accuracy on random samples, and the captions with a CLIP similarity. Most of the process was done on a single CPU node, with the exception of the deep learning models which were run on two GPUs.

Q24. **If the dataset is a sample from a larger set, what was the sampling strategy (e.g., deterministic, probabilistic with specific sampling probabilities)?**

The dataset is a sample from LAION. The URLs were chosen based on a list of curated fashion retailers domains, selected for the quality of their images and their use of simple and complex images to showcase a product.

Q25. **Who was involved in the data collection process (e.g., students, crowdworkers, contractors) and how were they compensated (e.g., how much were crowdworkers paid)?**

The authors are the only persons involved in this data collection process.

Q26. **Over what timeframe was the data collected? Does this timeframe match the creation timeframe of the data associated with the instances (e.g., recent crawl of old news articles)?** *If not, please describe the timeframe in which the data associated with the instances was created.*

The data was collected from LAION and annotated at the beginning of 2023. This timeframe does not match the timeframe associated with the instances. The LAION-5B dataset has been created between September 2021 and January 2022, based on CommonCrawl. CommonCrawl itself is a collection of webpages started in 2008. However, it is impossible to know for certain how far the data stretches, as the websites might include older pictures.

Q27. **Were any ethical review processes conducted (e.g., by an institutional review board)?** *If so, please provide a description of these review processes, including the outcomes, as well as a link or other access point to any supporting documentation.*

The dataset is currently under review.

Q28. **Does the dataset relate to people?** *If not, you may skip the remaining questions in this section.*

The dataset contains some images of fashion models, but it does not contain any personal data and focuses on objects.

Q29. **Did you collect the data from the individuals in question directly, or obtain it via third parties or other sources (e.g., websites)?**

No, we obtained it from LAION-5B.

Q30. **Were the individuals in question notified about the data collection?** *If so, please describe (or show with screenshots or other information) how notice was provided, and provide a link or other access point to, or otherwise reproduce, the exact language of the notification itself.*

Please refer to LAION-5B.

Q31. **Did the individuals in question consent to the collection and use of their data?** *If so, please describe (or show with screenshots or other information) how consent was requested and provided, and provide a link or other access point to, or otherwise reproduce, the exact language to which the individuals consented.*

Please refer to LAION-5B.

Q32. **If consent was obtained, were the consenting individuals provided with a mechanism to revoke their consent in the future or for certain uses?** *If so, please provide a description, as well as a link or other access point to the mechanism (if appropriate).*

Please refer to LAION-5B.

Q33. **Has an analysis of the potential impact of the dataset and its use on data subjects (e.g., a data protection impact analysis) been conducted?** *If so, please provide a description of this analysis, including the outcomes, as well as a link or other access point to any supporting documentation.*

This dataset and LAION 5B have been filtered using CLIP-based models. They inherit various biases contained in their original training set. Furthermore, the selected domains in this work only represent European and American fashion brands, and do not provide

Q34. **Any other comments ?**

No.

## C.4 PREPROCESSING / CLEANING / LABELING

Q35. **Was any preprocessing/cleaning/labeling of the data done (e.g., discretization or bucketing, tokenization, part-of-speech tagging, SIFT feature extraction, removal of instances, processing of missing values)?** *If so, please provide a description. If not, you may skip the remainder of the questions in this section.*

We started with a list of fashion domains with images of good quality, and extracted the corresponding images from LAION. We then trained a first classifier with an active learning procedure to classify the complexity of the obtained images. A second classifier was trained in the same way to classify the categories of the simple images, and captions were added using BLIP2-FlanT5XL.

We extracted product identifiers from the URLs, and kept products that were represented at least in a simple and a complex images. The discarded images, and those for which we couldn't extract any identifiers, are used as distractors.

We used LSH and KNN indices to remove duplicates among products, and between the products and the distractors in the validation and test sets.

Please refer to Appendix. A.2 for additional details.

Q36. **Was the "raw" data saved in addition to the preprocessed/cleaned/labeled data (e.g., to support unanticipated future uses)?** *If so, please provide a link or other access point to the "raw" data.*

The raw data is LAION-5B. It is available through LAION's HuggingFace pages.

Q37. **Is the software used to preprocess/clean/label the instances available?** *If so, please provide a link or other access point.*

No, apart from img2dataset that we used to download the images. Many critical parts in the process were manually supervised, such as extracting product identifiers for each domain, labeling during the active learning process, and checking the duplicates returned by the similarity search.

Q38. **Any other comments ?**

No.

## C.5 USES

Q39. **Has the dataset been used for any tasks already?** *If so, please provide a description.*

This is the first time that the LRVS-F dataset is used. We use it to study the Referred Visual Search task. The goal of this task is to retrieve a specific object among a large database of distractors given a complex image and additional referring information (category or text).

Q40. **Is there a repository that links to any or all papers or systems that use the dataset?** *If so, please provide a link or other access point.*

No.

Q41. **What (other) tasks could the dataset be used for?**

The dataset could be used for other fashion-related tasks, like fashion generation or virtual try-on.

Q42. **Is there anything about the composition of the dataset or the way it was collected and preprocessed/cleaned/labeled that might impact future uses?** *For example, is there anything that a future user might need to know to avoid uses that could result in unfair treatment of individuals or groups (e.g., stereotyping, quality of service issues) or other undesirable harms (e.g., financial harms, legal risks) If so, please provide a description. Is there anything a future user could do to mitigate these undesirable harms?*

Our dataset only contains large European and American fashion retailers. As such, it does not reflect the diversity of fashion cultures across the globe, and future users should not expect it to generalize to other geographical areas or specific localities.

Q43. **Are there tasks for which the dataset should not be used?** *If so, please provide a description.*

This dataset is for research purpose only, and contains biases. We warn any user against using it as-is outside of this context, and emphasize that results obtained on this dataset cannot be expected to generalize to any culture without proper bias study.

Q44. **Any other comments?**

As the images still belong to their respective owner, we only release this dataset for research purpose. We encourage anyone willing to use the images for commercial use to verify their copyright state with their respective rightholders.

Furthermore, we encourage users to respect opt-out policies, through the use of dedicated tools like img2dataset and SpawningAI.

## C.6 DISTRIBUTION

Q45. **Will the dataset be distributed to third parties outside of the entity (e.g., company, institution, organization) on behalf of which the dataset was created?** *If so, please provide a description.*

Yes, the dataset is open-source and freely accessible.

Q46. **How will the dataset be distributed (e.g., tarball on website, API, GitHub)?** *Does the dataset have a digital object identifier (DOI)?*

The dataset will be available as a collection of parquet files containing the necessary metadata. It will have a DOI.

Q47. **When will the dataset be distributed?**

It is already available.

Q48. **Will the dataset be distributed under a copyright or other intellectual property (IP) license, and/or under applicable terms of use (ToU)?** *If so, please describe this license and/or ToU, and provide a link or other access point to, or otherwise reproduce, any relevant licensing terms or ToU, as well as any fees associated with these restrictions.*

We release our data under the MIT license.

Q49. **Have any third parties imposed IP-based or other restrictions on the data associated with the instances?** *If so, please describe these restrictions, and provide a link or other access point to, or otherwise reproduce, any relevant licensing terms, as well as any fees associated with these restrictions.*

We only own the synthetic metadata that we release. The attributes of the dataset that originate from LAION-5B belong to LAION and are distributed under a CC-BY 4.0 license. We do not own the copyright of the images and original alt texts.

Q50. **Do any export controls or other regulatory restrictions apply to the dataset or to individual instances?** *If so, please describe these restrictions, and provide a link or other access point to, or otherwise reproduce, any supporting documentation.*

No.

Q51. **Any other comments ?**

No.

## C.7 MAINTENANCE

Q52. **Who will be supporting/hosting/maintaining the dataset?**

The dataset will be hosted at [Anonymized].

Q53. **How can the owner/curator/manager of the dataset be contacted (e.g., email address)?**

[Anonymized]

Q54. **Is there an erratum?** *If so, please provide a link or other access point*

There is no erratum as this is the initial release. If need be, we will update the dataset page on [Anonymized].

Q55. **Will the dataset be updated (e.g., to correct labeling errors, add new instances, delete instances)?** *If so, please describe how often, by whom, and how updates will be communicated to users (e.g., mailing list, GitHub)?*

We will not update the dataset, as it contains a benchmark and we want the results to stay comparable across time.

Q56. **If the dataset relates to people, are there applicable limits on the retention of the data associated with the instances (e.g., were individuals in question told that their data would be retained for a fixed period of time and then deleted)?** *If so, please describe these limits and explain how they will be enforced.*

The dataset does not relate to people. It does not contain personal or private information.

Q57. **Will older versions of the dataset continue to be supported/hosted/maintained?** *If so, please describe how. If not, please describe how its obsolescence will be communicated to users.*

There is currently no older version of this dataset. If changes must be made, the updates will be applied on the hosting page but history of changes will stay available.

Q58. **If others want to extend/augment/build on/contribute to the dataset, is there a mechanism for them to do so?** *If so, please provide a description. Will these contributions be validated/verified? If so, please describe how. If not, why not? Is there a process for communicating/distributing these contributions to other users? If so, please provide a description.*

We do not plan on supporting extensions to this dataset as it is intended to be a benchmark and results must stay comparable across time. However we do encourage the creation of new similar datasets across new verticals, to extend the field of Referred Visual Search.

Q59. **Any other comments ?**

No.

