# OpenReview forum: "Simplifying Referred Visual Search with Conditional Contrastive Learning"
_ICLR.cc/2024/Conference — Submitted to ICLR 2024_

### Official Review · Reviewer_sg3G · 2023-10-31

**Soundness:** 3 good
**Presentation:** 3 good
**Contribution:** 3 good
**Rating:** 5
**Confidence:** 5

**Summary:**

This paper proposes a new task called referred visual search. A new dataset is also created to achieve this task. This paper uses contrastive learning for extracting referred embeddings. Experiments achieve promising results in different tasks in LRVS-F dataset.

**Strengths:**

1. This paper proposes a challenging task for image similarity search in the context of fashion. A new dataset is also proposed at the same time.
2. Conditional embedding is properly used to achieve this task. Experiments demonstrate the effectiveness of the method.
3. This paper can be a baseline to do more relevant work.

**Weaknesses:**

1.This task is similar to composed image retrieval. Composed image retrieval aims to find the target image based on the reference image and text description. I have some doubts about the contribution and meaning of the task.
2.The structure of the model is simple. It lacks innovation. The description of the model is not specific enough. Contrastive learning is often used in the task of composed image retrieval, so it is not an innovative method.
3.Experiments are basically a comparison with other models, but the ablation experiment of your own model and visualization is lacking.

**Questions:**

1. You say it extracts referred embedding using weakly-supervised training. Why it is a weakly-supervised training?
2. In fig2, it shares weight between the two vision transformers. In this model, it is whether all parameters are shared or only parts of the parameters are shared. And I want to know why to share the weight.
3. In table1 and table2, you say you report bootstrapped means and standards deviations for 0K distractors, but I don’t see the result of the 0K distractors. In addition, your model is similar to some models that used in the task of composed image retrieval. FashionIQ is also a dataset about clothes. Do you try this dataset using your method?

---

> ### Author Response · Authors · 2023-11-14
>
> We thank the reviewer for their feedback and address their main concerns below :
>
> > **W1.** This task is similar to composed image retrieval. Composed image retrieval aims to find the target image based on the reference image and text description. I have some doubts about the contribution and meaning of the task.
>
> Our RVS task is different from Composed Image Retrieval in that in RVS the object in the query image and in the target image are exactly the same. The use case is retrieving that object from a catalog, given a picture that contains many other different objects. The goal of the model is thus to be able to extract the correct information from the query image. In contrast, in CIR, the target object is not the same as any of the objects present in the query image. The goal of a CIR model is to modify the query image such that it hallucinates an object that is similar to what is in the catalog. Both tasks have relevant applications, but they are sufficiently different to be addressed with different methods. Indeed, we show that models made for CIR perform poorly in RVS, which is intuitive: they are not designed to extract the relevant information from the image, but to modify it to hallucinate an object that matches the textual description.
>
> > **W2.** The structure of the model is simple. It lacks innovation. The description of the model is not specific enough. Contrastive learning is often used in the task of composed image retrieval, so it is not an innovative method.
>
> It works, and the goal of the paper was explicitly to show that complex pipelines are not needed to reach good performances. Simplification of complex tasks so they become accessible to more people and robust has often more impact than a complex system prone to brittleness
>
> > **W3.** Experiments are basically a comparison with other models, but the ablation experiment of your own model and visualization is lacking.
>
> Due to space limitations, we rejected 2 ablations in the Appendix related to our method: how to condition (symmetric/asymmetric) and where to condition (insertion depth), with plenty of retrieval results, attention visualization and PCA, as well as failure cases. Squeezing them into the main paper would force us to cut related work or experimental descriptions.
>
> > **Q1.** You say it extracts referred embedding using weakly-supervised training. Why it is a weakly-supervised training?
>
> We call it weakly-supervised because we don’t have access to localized information (such as bounding boxes) about the object of interest. In the computer vision literature, this is referred to as weak supervision, in contrast to having access to bounding box or segmentation maps which are usually referred to as supervised methods.
>
> > **Q2.** In fig2, it shares weight between the two vision transformers. In this model, it is whether all parameters are shared or only parts of the parameters are shared. And I want to know why to share the weight.
>
> Our method uses a single model to process both unconditional and conditional images, so the weights are the same. It is less memory intensive to have the same model and we did not find any negative impact.
>
> > **Q3.** In table1 and table2, you say you report bootstrapped means and standards deviations for 0K distractors, but I don’t see the result of the 0K distractors. In addition, your model is similar to some models that used in the task of composed image retrieval. FashionIQ is also a dataset about clothes. Do you try this dataset using your method?
>
> Thanks for pointing out the mistake in the table caption, we removed the results with 0 distractors to make the table more legible. The results were not really interesting as performances in this very simple setting tend to saturate.
>
> We are currently running experiments on FashionIQ and will report the results here. However please note that this is a different task and we do not expect state-of-the-art results.

---

> > ### Author Response · Authors · 2023-11-23
> >
> > As announced, we ran additional experiments on FashionIQ and posted the results as an [Official Comment](https://openreview.net/forum?id=Z8uFGTNXIF&noteId=47CMQgCUha).

---

### Official Review · Reviewer_vVJq · 2023-11-03

**Soundness:** 3 good
**Presentation:** 3 good
**Contribution:** 3 good
**Rating:** 5
**Confidence:** 4

**Summary:**

This paper proposes a new task, named Referred Visual Search (RVS). It aims to search the specific part under the condition of category. The new task sounds good. The authors also introduce a corresponding dataset and framework. Some experimental results look good.

**Strengths:**

## Strengths

1. The writing is good. It is easy to follow this paper.

2. The motivation sounds reasonable.

3. Some experimental results look good.

**Weaknesses:**

## Weaknesses

1. It may be not appropriate to use the entire image (even given the conditions) to search for an part area, such as pants.

2. Why not crop the part area according to the given condition and then use it to search?

3. How to collect the LAION-RVS-Fashion dataset? How to ensure the accuracy of the labels? The original labels are not clear.

4. The main content of this paper has 10 pages.

**Questions:**

Please see above.

---

> ### Author Response · Authors · 2023-11-14
>
> We thank the reviewer for their feedback and address their main concerns below:
>
> > **W1.** It may be not appropriate to use the entire image (even given the conditions) to search for an part area, such as pants.
>
> While surprising, the point of this paper is to demonstrate that it is currently the best performing approach (see results in Table 2).
>
> > **W2.** Why not crop the part area according to the given condition and then use it to search?
>
> Very good point. Maybe it wasn’t stressed enough in the paper but it is exactly our baseline with GroundingDINO + ViT (Section 5.2, §Detection-based Baseline, and Table 1).
>
> > **W3.** How to collect the LAION-RVS-Fashion dataset? How to ensure the accuracy of the labels? The original labels are not clear.
>
> We will add a script for direct downloading to a public repository (using img2dataset on the provided parquet files).
>
> We are very confident on the labels since the tuples are made from a limited number of shop websites using rule-based parsing.
>
> > **W4.** The main content of this paper has 10 pages.
>
> We followed the author guidelines stating that ethics and reproducibility statements are “not counted against the maximum page limit”. See https://iclr.cc/Conferences/2024/AuthorGuide .

---

### Official Review · Reviewer_F8wp · 2023-11-04

**Soundness:** 3 good
**Presentation:** 4 excellent
**Contribution:** 3 good
**Rating:** 6
**Confidence:** 3

**Summary:**

This paper aims at fashion retrieval conditioned on images and texts. Particularly, the text conditions can be categories and captions. This task is tackled via learning the joint embedding of texts and images, similar to conventional multi-modal metric learning methods. A dataset that is extracted from the publicly available dataset LAION-5B is constructed to validate the proposed method.

**Strengths:**

1. The proposed dataset can facilitate research on multi-modal fashion retrieval.
2. Extensive experiments have been conducted to provide insightful information on this task.
3. The writing of this paper is excellent and easy to follow.

**Weaknesses:**

1. Although this paper claims its target task is new, I still consider it to belong to multi-modal fashion retrieval. That is, one can include classes, attributes, captions, or even negative prompts in the textual conditions, and then leverage LLMs to process them uniformly.
2. The failure cases suggest image features are dominant, and hence the proposed method or the task might not be as convenient as it claims. For example, what if the user wants to find clothes with similar styles but different colors, or of the same brand? Moreover, the text conditions seem rather simple, so whether the proposed method can handle fine-grained queries is unclear.
3. The comparison between the proposed method and SOTAs might be unfair, e.g., ASEN is implemented partially and it only uses attributes. Other baselines with similar architectures, like FashionBert should be considered as well. Besides, how is the performance of the proposed method on other fashion retrieval benchmarks?

**Questions:**

Please refer to the weakness part. I will adjust my score according if my concerns can be addressed.

---

> ### Author Response · Authors · 2023-11-14
>
> We thank the reviewer for their feedback, and address their main concerns below :
>
> > **W1.** Although this paper claims its target task is new, I still consider it to belong to multi-modal fashion retrieval. That is, one can include classes, attributes, captions, or even negative prompts in the textual conditions, and then leverage LLMs to process them uniformly.
>
> We agree that our RVS task belongs to multi-modal fashion retrieval. However, it has not been tackled in the literature, contrary to text-to-image or image-to-text retrieval and composed image retrieval for which known datasets and benchmarks exist.
>
> For text, we do use a T5-XL to produce our embeddings and enable free-form conditioning. We find experimentally that using a single word (~category) as input works well (Fig. 9 in the appendix) even though BLIP2 captions are longer.
>
> > **W2.** The failure cases suggest image features are dominant, and hence the proposed method or the task might not be as convenient as it claims. For example, what if the user wants to find clothes with similar styles but different colors, or of the same brand? Moreover, the text conditions seem rather simple, so whether the proposed method can handle fine-grained queries is unclear.
>
> Composed Image Retrieval is a different task: in our task we need to retrieve the exact garment depicted on the image. We think that paying more attention to the image is a good property, text is only here to designate an object in the scene. Unifying RVS & CIR is indeed an interesting direction, as stated in our conclusion.
>
> Regarding fine-grained text conditioning, we are currently running experiments on a Fashion CIR dataset (FashionIQ) and will report the results here. However please note that this is a different task and we do not expect state-of-the-art results.
>
>
> > **W3.** The comparison between the proposed method and SOTAs might be unfair, e.g., ASEN is implemented partially and it only uses attributes. Other baselines with similar architectures, like FashionBert should be considered as well. Besides, how is the performance of the proposed method on other fashion retrieval benchmarks?
>
> We fully trained ASEN using the author’s implementation and reported the best obtained results (attained during stage 1). The results obtained with stage 2 are 14.6 R@1 against2M. We believe this is due to the asymmetry of this task in which the gallery is composed of simple images containing only a single object that break key assumptions of ASEN. This result highlights the difference of our RVS task compared to other retrieval tasks for which ASEN was designed.
>
> Since FashionBERT/UNITER are designed for cross-modal retrieval, it is unclear how they could be used for conditional retrieval. A naive approach would result in a quadratic cost with respect to the number of products which is prohibitive here (in our case, 2k queries times 2M distractors results in 4B forward passes which would take more than a year of compute at 100 forward passes per second).
>
> To our knowledge, there exists no other public RVS benchmark, where the task is to conditionally extract an embedding of a subpart of the image. Our method has not been designed for unconditional Image to Text retrieval or Text to Image retrieval tasks.
>
> We are currently running experiments on a Fashion CIR dataset (FashionIQ) and will report the results here. However please note that this is a different task and we do not expect state-of-the-art results.

---

> > ### Author Response · Authors · 2023-11-23
> >
> > As announced, we ran additional experiments on FashionIQ and posted the results as an [Official Comment](https://openreview.net/forum?id=Z8uFGTNXIF&noteId=47CMQgCUha).

---

### Author Response · Authors · 2023-11-23
**Results on FashionIQ**

We report the results of our experiments on FashionIQ, as mentioned in our answers to reviewers F8wp and sg3G.

We initialize a CondViT-B/32 from a checkpoint trained on LRVSF with textual conditioning and fine-tune it on FashionIQ.
Because of the very limited size of the dataset (18K training triplets), we adopt the following modifications:
- We add a trainable 2-layers MLP applied on the text embedding to account for the distribution shift between the initial BLIP2 captions and FashionIQ’s annotations.
- We insert the conditioning token at layer 6, and only train subsequent layers.
- We evaluate a weight-space average of 5 runs.

We report usual metrics for this dataset: Recall at 10 and 50 on the validation set for each clothing category, and averages.

|   Method | dress R@10 | dress R@50 | shirt R@10 | shirt R@50 | toptee R@10 | toptee R@50 | average R@10 | average R@50 | average |
| -------: | :--------: | :--------: | :--------: | :--------: | :---------: | :---------: | :----------: | :----------: | :-----: |
|    CoSMo |   25.64    |    50.30    |    24.90    |   49.18    |    29.21    |    57.46    |    26.58     |    52.31     |  39.45  |
|     Ours |    29.60    |   56.47    |    26.10    |   49.31    |    32.48    |    59.46    |     29.40     |    55.08     |  42.24  |
| CLIP4CIR |   33.81    |    59.40    |   39.99    |   60.45    |    41.41    |    65.37    |    38.32     |    61.74     |  50.03  |

As expected, the ranking of the methods on FashionIQ is different than that of LRVSF which highlights how different the tasks are. Indeed, while our method tailored for referred visual search performs best on LRVSF it achieves only 2nd place on FashionIQ among the methods tested in our paper. Conversely, CLIP4CIR which achieves best on FashionIQ for which it was designed is only 3rd on LRVSF, performing similarly to our smallest baseline.

These results also hint that our method can be adapted to more complex conditioning (ie textures, colors or shape modifications).

---

### Meta-Review · Area_Chair_ETQf · 2023-12-05

**Metareview:**

While the reviewers recognize the merits of the work (useful dataset and baseline, conditional embedding/retrieval), they also identify weaknesses in the method's novelty (e.g. focus on the well-known contrastive learning), validity/practicality, and experiments (fairness, lack of ablations). The paper received borderline scores and no reviewer strongly advocated for acceptance.

**Justification For Why Not Higher Score:**

No scores above 6 (only one 6)

**Justification For Why Not Lower Score:**

n/a

---

### Decision · Program_Chairs · 2024-01-16

Reject